# Observation of solid–solid transitions in 3D crystals of colloidal superballs

Janne-Mieke Meijer[1,†], Antara Pal[1,†], Samia Ouhajji[1], Henk N.W. Lekkerkerker[1], Albert P. Philipse[1] & Andrei V. Petukhov[1,2]

Self-organization in anisotropic colloidal suspensions leads to a fascinating range of crystal and liquid crystal phases induced by shape alone. Simulations predict the phase behaviour of a plethora of shapes while experimental realization often lags behind. Here, we present the experimental phase behaviour of superball particles with a shape in between that of a sphere and a cube. In particular, we observe the formation of a plastic crystal phase with translational order and orientational disorder, and the subsequent transformation into rhombohedral crystals. Moreover, we uncover that the phase behaviour is richer than predicted, as we find two distinct rhombohedral crystals with different stacking variants, namely hollow-site and bridge-site stacking. In addition, for slightly softer interactions we observe a solid–solid transition between the two. Our investigation brings us one step closer to ultimately controlling the experimental self-assembly of superballs into functional materials, such as photonic crystals.

[1] Van 't Hoff Laboratory for Physical and Colloid Chemistry, Debye Institute for Nanomaterials Science, Utrecht University, 3584 CH Utrecht, The Netherlands. [2] Laboratory of Physical Chemistry, Department of Chemical Engineering and Chemistry, Eindhoven University of Technology, 5600 MB Eindhoven, The Netherlands. † Present addresses: Department of Physics, University of Konstanz, D-78457 Konstanz, Germany (J.-M.M.); Physical Chemistry, Department of Chemistry, Lund University, SE-22100 Lund, Sweden (A.P.). Correspondence and requests for materials should be addressed to J.-M.M. (email: janne-mieke.meijer@uni-konstanz.de) or to A.V.P. (email: a.v.petukhov@uu.nl).

Self-organization in colloidal suspensions leads to a fascinating range of crystal and liquid crystal phases[1–3] and is a promising route for the fabrication of novel materials with designed (optical) properties[4–7]. Traditionally the focus has been on the self-assembled structures of colloids with basic shapes (spheres, rods and platelets)[8,9]. Recently due to advances in synthesis[1,7], both experiments and simulations have focused on exploring the self-assembled structures of various complex anisotropic particles[10], such as dumbbells[5,6], cubes[11–14], polyhedra[15,16], octapods[17] and even co-crystals of complementary shapes[18]. Not limited by the experimental restrictions, simulations have gone even further and have already investigated the phase behaviour and dense packings for an even larger variety of anisotropic shapes[2,3,19–21]. One of the fascinating self-assembled phases is the plastic crystal (PC) which possesses translational order but orientational disorder and hence is the opposite of a liquid crystal. The PC phase is well known for molecular systems[22–24], but for colloidal suspensions this phase is rather rare and has only been reported in a few three-dimensional (3D) studies[25–29].

One of the anisotropic colloidal shapes that have recently become experimentally available is the superball[11,30]. The superball family describes the shape that smoothly interpolates between a sphere and a cube (Fig. 1a) and can be represented by

$$\left|2\frac{x}{L}\right|^m + \left|2\frac{y}{L}\right|^m + \left|2\frac{z}{L}\right|^m \leq 1, \tag{1}$$

where $m$ is the shape parameter and $L$ is the face-to-face-length of the superballs. Using simulations the optimal packings and subsequently the phase behaviour has been studied in both two-dimensional (2D) and 3D[31–36]. These studies found that due

to the rounded corners of the cube-like shape, a close-packed rhombohedral crystal (RC) is formed with $C_1$-lattice structure, which for $2.3 < m < 5$ is preceded by a PC phase. Experimentally, the phase behaviour has been determined in 2D[37], but in 3D, confirmation[11] exists only for the densest packings for $m > 5$ and full phase behaviour studies are still lacking.

In this communication we focus on the experimental self-assembled structures of a system of silica superball colloids[30,38–41] and show that by tuning the shape, size and interactions we can study the phase behaviour for $m < 5$, where the enriched phase behaviour is predicted[34]. Using a combination of confocal microscopy and high resolution X-ray scattering we find an even richer phase behaviour than predicted, as three distinct crystal phases are uncovered. Particularly, we find a solid–solid transition from a PC phase into two different RC phases, one with hollow-site stacking while the other possesses bridge-site stacking. We further investigate how the phase diagram depends on the exact superball shape and osmotic pressure in the system and additionally find that a slight softness causes a second solid–solid transition from the hollow-site to bridge-site RC phase at high pressures. Our investigation brings us one step closer to ultimately controlling the experimental superball self-assembly into functional materials, such as photonic crystals.

## Results

**Superball crystal phases.** To investigate the effect of superball shape on the experimental 3D phase behaviour we employed hollow silica superballs[30,38–40] with values of $m$ between 2.9–3.6 and sizes in the micron range (Fig. 1b and Supplementary Table 1). For *in situ* imaging of the superball assemblies, we used confocal laser scanning microscopy (CLSM) that allows direct visualization of the local superball positions and orientations in

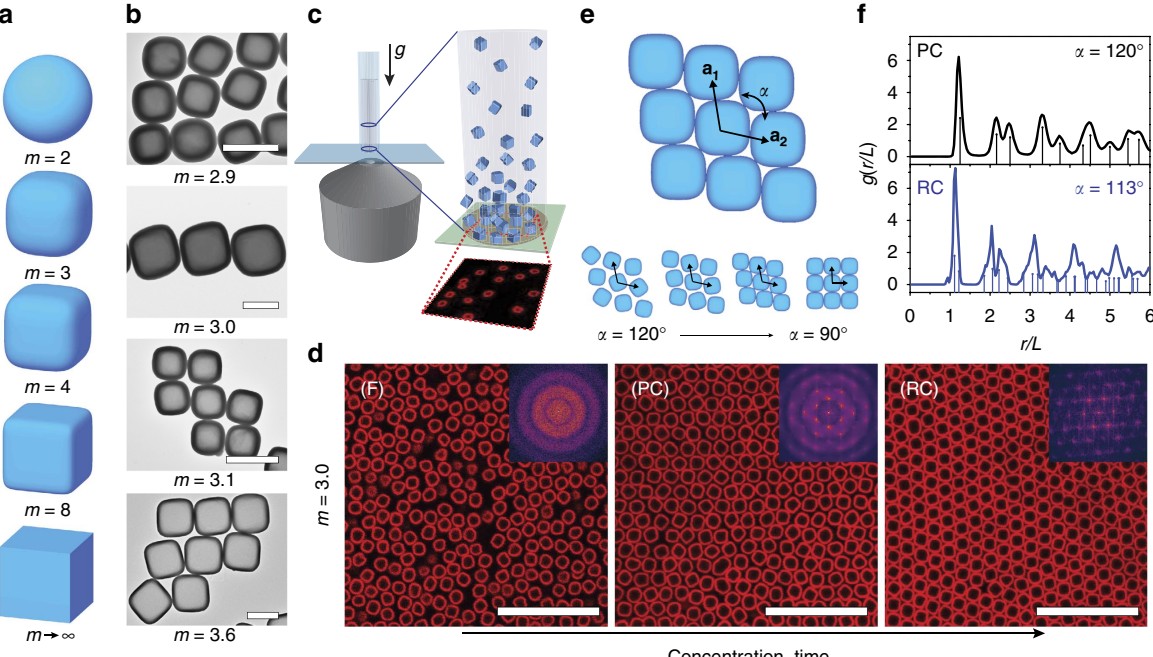

**Figure 1 | The assembly of hollow superball colloids into different phases.** (**a**) Schematic representation of superball particles of which the corner roundness is described by the shape parameter $m$. (**b**) Transmission electron microscopy images of several of the hollow silica superball particles. (**c**) CLSM set-up of sedimentation experiment of superball particles onto a cover slide. (**d**) CLSM images obtained just above the glass wall of fluorescent superballs with $m = 3.0$ showing the structural transition over time from a disordered fluid phase (F) to a PC and finally into a RC lattice consisting of stacked RC planes (see Supplementary Movie 1 for particle dynamics and Supplementary Fig. 1 for cross sections). (Insets) Fourier transform of time series of the respective phase. (**e**) Schematic representation of an RC plane described by lattice vectors $\mathbf{a_1}$ and $\mathbf{a_2}$ and their angle, $\alpha$. The complete transition from a hexagonal to square lattice is described by a change in $\alpha$ from 120° to 90°. (**f**) Radial distribution functions $g(r/L)$ extracted from the particle positions in the aligned lattice planes together with the expected positions for different rhombohedral lattice angles $\alpha$. Scale bars are (**b**) 1 µm (**d**) 10 µm.

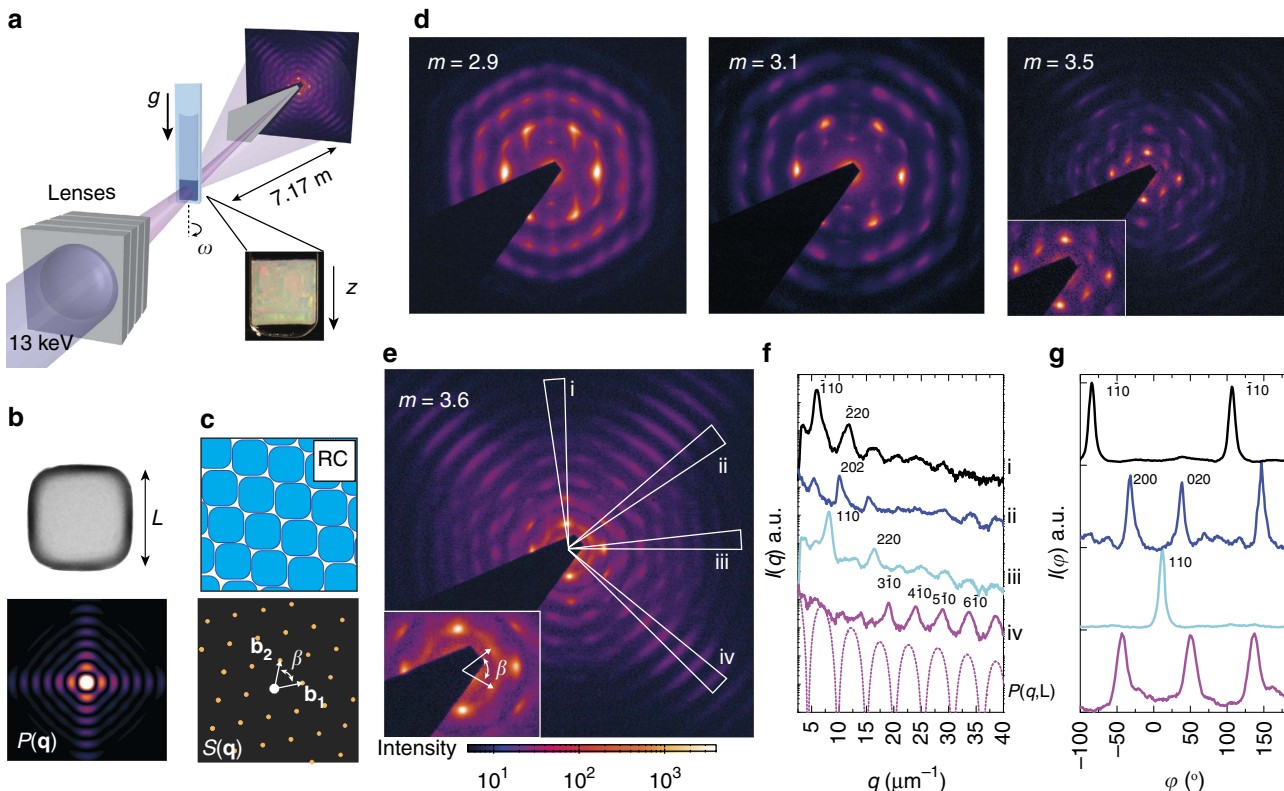

**Figure 2 | The effect of shape on the 3D crystals of superball colloids.** (**a**) Schematic set-up of the μrad-SAXS set-up used to investigate the 3D structure of the superball sedimentary crystals. (Inset) Photograph of a sample displaying optical Bragg reflections. (**b**) Transmission electron microscopy image of a hollow superball particle with $m = 3.6$ and numerically calculated 2D form factor $P(\mathbf{q})$ for the shown orientation. For the anisotropic hollow superballs ($m > 2$) $P(\mathbf{q})$ is also anisotropic, showing a distinct square pattern. (**c**) Schematic representation of a close-packed rhombohedral lattice (RC) aligned to the capillary wall and 2D structure factor $S(\mathbf{q})$ for $m = 3.6$. $S(\mathbf{q})$ is described by reciprocal space vectors $\mathbf{b_1}$ and $\mathbf{b_2}$ and their angle, $\beta$, with a range of $60° \rightarrow 90°$, corresponding to the real space RC angle $\alpha$, with a range of $120° \rightarrow 90$. (**d,e**) Selected experimental 2D μrad-SAXS patterns of the superball sediments showing mono-crystalline regions with RC symmetry. (**f,g**) Extracted profiles along the $q$-wedges and azimuthal rings of the 2D pattern in **e**. (**f**) $I(q)$ profiles along i–iv $q$-wedges and (**g**) normalized $I(\varphi)$ profiles of the different $q_{hkl}$ diffraction peaks, the curves are offset for clarity.

the crystal lattices. These results were corroborated with synchrotron small-angle X-ray scattering with microradian resolution[10] (μrad-SAXS) that allows us to determine the overall crystal symmetry and lattice parameters of the colloidal crystals in solution. In the CLSM experiment we employed a 1.0 wt% dispersion in ethanol of fluorescently labelled superball particles with $L = 1.273 \pm 0.051\,\mu m$ and $m = 3.0$. The Debye screening length was estimated to be $\kappa^{-1} \sim 50\,nm$, leading to slightly 'softer' interactions when separation between particle surfaces reduces to a few per cent of their size. In the following the particles in these samples will be referred to as soft to distinguish them from other samples where the Debye length was reduced to $\sim 4\,nm$ by adding salt. The settling of the dispersion to the bottom cover slide of the sample cell was imaged over time (Fig. 1c). The superball assemblies formed by this sedimentation procedure consist of three different phases (Fig. 1d, see also Supplementary Fig. 1 and Supplementary Movie 1): a fluid phase (F) in which the particles diffuse freely and have random orientations, a PC phase with translational order and orientational disorder, and a crystal with rhombohedral symmetry (RC) with translational and orientational order. In both crystal phases the lattice planes are aligned to the cover slide, but only in the rhombic crystal the superball faces were found to anchor to the wall. Our 3D observations are similar to the previously observed 2D crystal–crystal transition[36,37] from a hexagonal rotator phase into the predicted thermodynamically stable structure for superdisks with $m > 2.57$ and are described by the angle variable $\Lambda_1$-lattice[32]. In addition, recent experiments

on silica superballs using convective assembly[38] or depletion interaction[40] also showed that the $\Lambda_1$-lattice is formed by superballs in 2D. Here we describe the RC planes based on the $\Lambda_1$-lattice with the vectors $\mathbf{a_1}$ and $\mathbf{a_2}$, with angle, $\alpha$. A change in $\alpha$ from $120°$ to $90°$ depicts a continuous transformation from a hexagonal to square lattice (Fig. 1e, for further details see Supplementary Note 1 and Supplementary Fig. 2).

The structural properties of the phases were investigated by computing the Fourier transform for a time series of CLSM images that clearly show the disorder, hexagonal order and rhombohedral order, respectively (insets of Fig. 1d). In addition, we extracted the 2D radial distribution function, $g(r/L)$, in the crystal lattice planes (Fig. 1f). The PC phase $g(r/L)$ peaks correspond to an RC plane with $\alpha = 120°$ and an interparticle spacing along the superball face-to-face direction, $d = 1.25L$. A superball can freely rotate in 3D when $d > L^{(3)}$, the length of the superball space diagonal[42] $L^{(3)} = \sqrt{3}L(1/3)^{1/m}$. For $m = 3.0$ we find $L^{(3)} = 1.2L$, confirming that the superballs are able to explore all orientations on their lattice site. For the RC phase good agreement is found with $\alpha = 113°$ and $d = 1.09L$, where $\alpha$ corresponds to the angle expected for the $\Lambda_1$-lattice with $m = 3.0$ (ref. 32).

**Rhombic crystal structure.** To investigate the crystal–crystal transitions in 3D, we studied the bulk sedimentary crystals of the soft fluorescent silica superballs with μrad-SAXS. In addition, to see the effect of truly 'hard' interactions as well as the effect

of shape, we also studied the sediments of unlabelled silica superballs with different $m$ values. These silica superballs were dispersed in an alkaline water solution with 6 mM salt that reduces the Debye length to $\kappa^{-1} \sim 4$ nm. The µrad-SAXS set-up consists of compound refractive lenses placed in front of the sample that focus the X-ray beam on a 2D detector placed 7.17 m behind the sample (Fig. 2a)[10]. The microradian resolution is crucial since the diffraction peaks appear at scattering angles of $\sim 0.005°$ for the largest colloids used here. The total measured scattered intensity $I(\mathbf{q})$ on the 2D detector is a product of the form factor $P(\mathbf{q})$ of the hollow superball shell and the structure factor $S(\mathbf{q})$, according to $I(\mathbf{q}) \propto P(\mathbf{q})S(\mathbf{q})$. For the anisotropic superball shells $P(\mathbf{q})$ depends on the orientation and the shell thickness with respect to the length ratio, $t/L$, of the superball (Fig. 2b). For an RC plane $S(\mathbf{q})$ can be described by the reciprocal space vectors $\mathbf{b_1}$ and $\mathbf{b_2}$ with angle $\beta$ (Fig. 2c) that are directly related to the real space vectors[43]. In 3D reciprocal space the scattering vector $\mathbf{q}$ of $S(\mathbf{q})$ is described by $\mathbf{q} = h\mathbf{b_1} + k\mathbf{b_2} + l\mathbf{b_3}$ (See Supplementary Note 1 and Supplementary Fig. 2).

The 2D µrad-SAXS patterns obtained from the different sediments were all found to be very anisotropic and show exceptionally sharp mono-crystalline Bragg peaks with an RC symmetry (Fig. 2d,e). The formation of mono-crystalline domains, which are larger than the diameter of X-ray beam $\sim 500$ µm, is attributed to alignment of the superball faces to the capillary wall. This is followed by subsequent nucleation of the crystals consisting of close-packed RC planes. The wall-anchoring was confirmed by measurements in round capillaries where poly-crystalline patterns were observed (Supplementary Fig. 3). With increasing $m$ the RC patterns seem to become more rhombohedral. This change in symmetry can be partly explained by a difference in the $P(\mathbf{q})$ fine structure due to the specific $m$ and relative shell thickness $t/L$ of the superballs (Supplementary Fig. 4) but this does not explain the change fully.

Therefore, the structural properties of the crystals were investigated by extracting the intensity profiles along the radial, $q$, and azimuthal, $\varphi$, direction of the diffraction peaks (Fig. 2f,g). The peaks were identified as the $hk0$ peaks originating from the rhombohedral lattice planes. The correspondence of the $I(q)$ profile (iv-wedge) to the calculated $P(q)$ profile along $L$, shows the high degree of orientational order of the superballs in the lattice (Fig. 2f). Due to the 2D nature of the mono-crystalline µrad-SAXS patterns, the structural angle $\beta$ can be extracted directly by measuring the angle between the $h00$ and $0k0$ peaks in the $\varphi$-profiles or by calculating it from the $q_{hk0}/q_{110}$ ratio (see Supplementary Note 1 and Supplementary Fig. 2). Moreover, from $\beta$ and the $q_{hkl}$ positions the effective in-plane particle spacing, $d$, in the crystals can be determined. For the superball particles with $m = 3.6$ we find good agreement between the experimental value $\beta = 70.6°$ and the closest-packing value $\beta = 70.2°$, as well as for the crystal spacing with $d = 1.286$ µm compared with $L = 1.266 \pm 0.027$ µm.

Figure 3a shows the extracted $\beta$ and $d/L$ for each of the investigated superball shapes $m$. Good agreement is found with the theoretical $\beta$ expected for a close-packed $\Lambda_1$-lattice of the same $m$. Moreover, $d/L \rightarrow 1$ for each $m$ showing that the RC planes are densely packed in all cases. Noteworthy is that both the soft fluorescent superballs with $m = 3.0$ and the differently shaped hard superballs all have formed close packed structures.

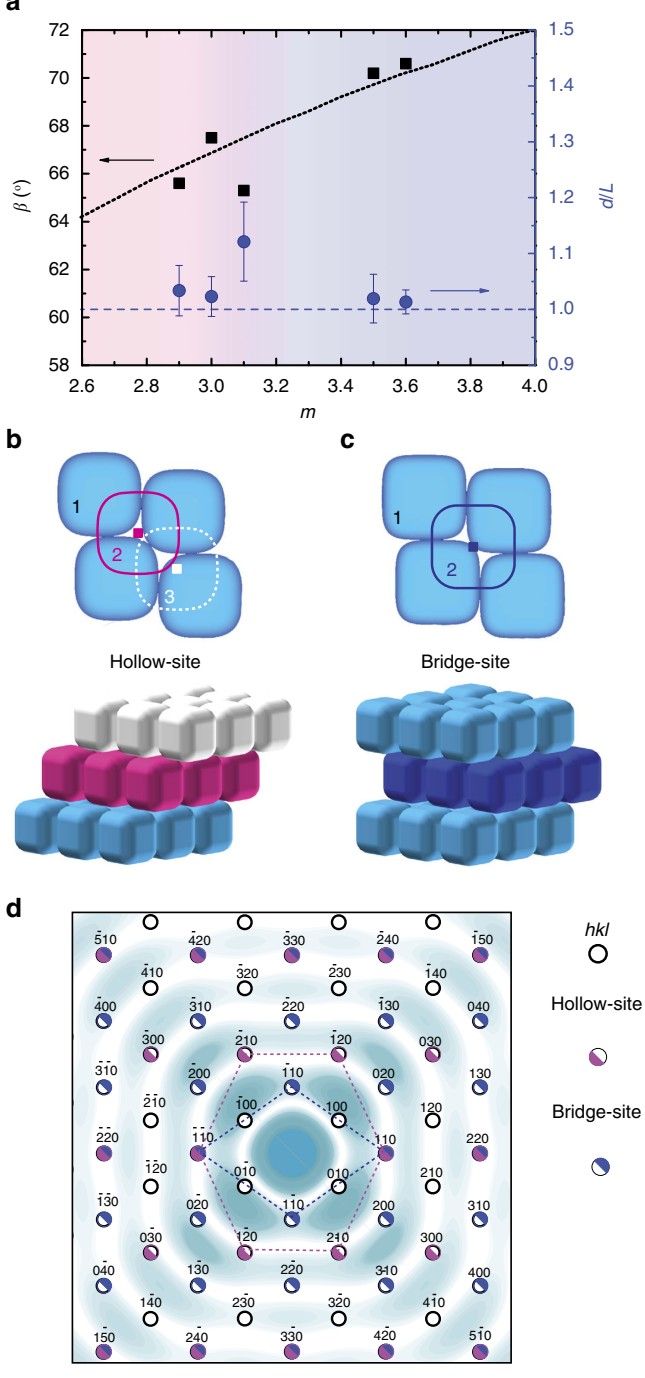

**Figure 3 | The superball shape changes the densely packed RC stacking sequence.** (**a**) For each of the superball shapes the structural angle, $\beta$, and ratio of the lattice spacing and the superball edge-to-edge length, $d/L$, were determined. The error bars of the latter represent the scaled standard deviation in superball size, $\sigma_L/L$. Good agreement is found with theoretical $\beta$ (black dotted line) expected for a close-packed ($d/L = 1$, blue dashed line) rhombic crystal (RC) planes. (**b,c**) Schematic illustrations of the position of successive close-packed RC planes that can be either positioned into (**b**) the hollow-site (HRC) or (**c**) the bridge-site (BRC) position and the resulting 3D structure for both stacking sequences. (**d**) Numerically calculated 2D patterns of $P(\mathbf{q})$ and $S(\mathbf{q})$ for a superball shell with $m = 3.6$ and shell thickness to length ratio, $t/L = 0.03$. Due to constructive and destructive interference between the RC planes, only specific $hkl$ peaks are allowed in the $S(\mathbf{q})$ pattern for each of the two stacking sequences, resulting in a hexagonal pattern for HRC or a rhombic pattern for BRC, that are enhanced by $P(\mathbf{q})$.

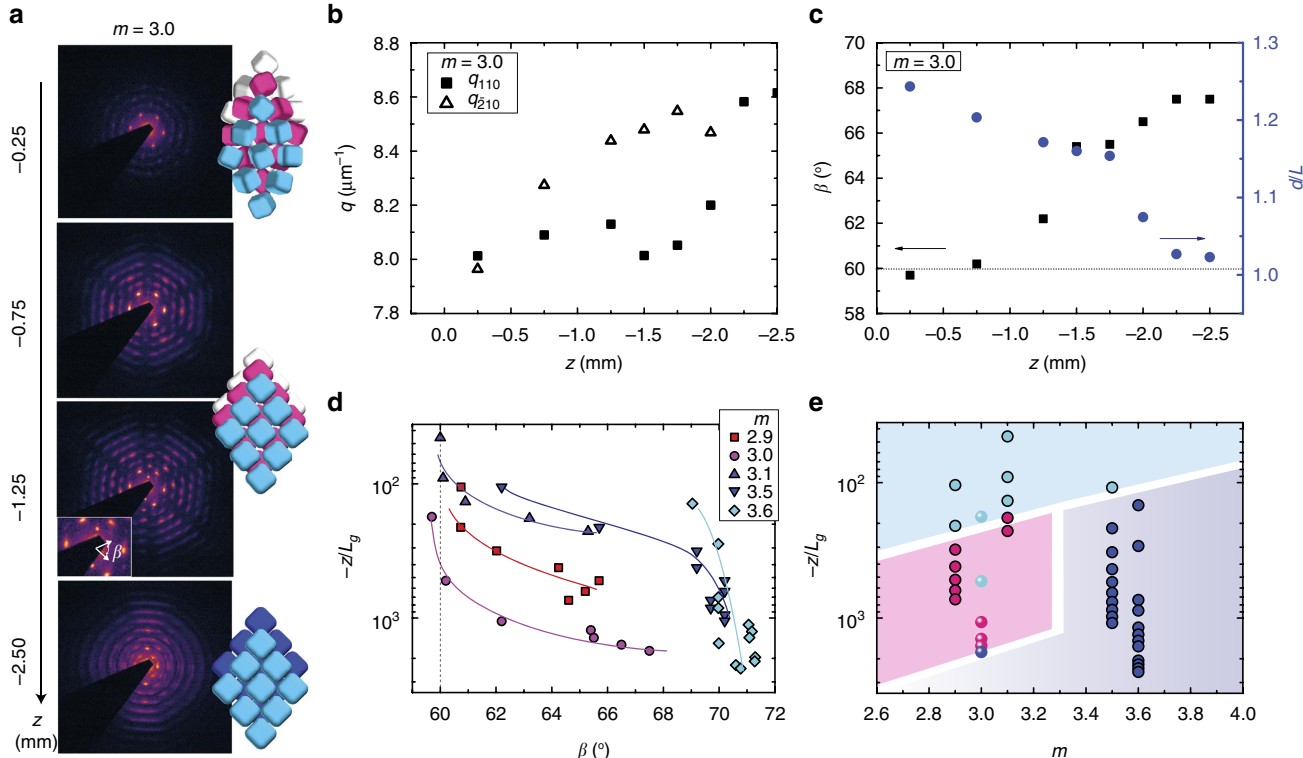

**Figure 4 | Different solid–solid transitions occur depending on osmotic pressure and superball shape.** (**a**) 2D µrad-SAXS patterns for the superballs with $m = 3.0$ of the crystalline sediment at different heights, $z$, in the capillary, together with schematic representation of the different crystal phases that are found; a plastic crystal (PC), a rhombic crystal with hollow-site stacking (HRC) and a rhombic crystal with bridge-site stacking (BRC). (**b**) $q$-positions of the two brightest $hk0$ peaks in the patterns of (**a**), indicating a structural transition. (**c**) The structural angle, $\beta$, and ratio of the lattice spacing and the superball edge-to-edge length, $d/L$, for the different heights in the sediment showing that the structure goes from a low density hexagonal ($\beta \sim 60°$) structure to a more densely packed rhombohedral ($\beta > 60°$) structure further down in the sediment. (**d**) Relative pressure given by $-z/L_g$ versus angle $\beta$ found for the different superballs, showing the presence of phase transitions in almost all sediments. Note that the $-z/L_g$ scale is reverted such that the top of the graph corresponds to the sediment top and the lines are a guides to the eye. (**e**) Experimental phase diagram established for the studied superball shapes, where a PC phase (light blue) transforms into a HRC phase (pink) or a BRC phase (dark blue) depending on the superball $m$. Circles with uniform filling are for hard superballs while particles with $m = 3.0$ have slightly softer interactions. The overall phase transitions are guides to the eye.

The full 3D structure was investigated by rotating the capillaries around their vertical axis by $\omega \pm 70°$, where $\omega = 0°$ corresponds to the capillary orientation in Fig. 2a. Surprisingly, with increasing shape parameter different $hkl$ peaks were observed in the 2D patterns (see Supplementary Fig. 5). On the basis of the observed peaks two distinct different stacking sequences of the RC planes were identified; namely hollow-site and bridge-site stacking, which will be referred to as HRC and BRC, respectively. Figure 3b,c show the schematic representation of the layer positions in the two stacking sequences together with a 3D representation. The formation of the two stacking sequences is dependent on the superball shape parameter $m$. For $m > 3.1$ we find HRC co-existing with a small amount of BRC, while for $m \geq 3.5$ only BRC stacking is observed. Furthermore, the different stacking sequences are also the cause of the different $hkl$ intensities in the µrad-SAXS patterns at $\omega = 0°$, that is, for $l = 0$. (Fig. 2d,e). Due to the constructive and destructive interference between the successive layers certain $hk0$ peaks are not allowed. At normal incidence a HRC structure will only show $hk0$ peaks that fulfil the condition that $(h-k)$ is divisible by 3, while for a BRC structure only $hk0$ peaks fulfilling the condition that $(h-k)$ is divisible by 2. Figure 3d shows the expected position of all the $hk0$ peaks of $S(q)$ on top of the numerically calculated $P(q)$ for the superballs with $m = 3.6$, showing a hexagonal pattern for the HRC structure and a rhombic pattern for the BRC structure. Interestingly, the $hk0$ peaks observed in Fig. 2e are

found to overlap with minima in $P(q)$, thereby confirming their structural origin.

**Solid–solid transitions.** To study the effect of osmotic pressure on the 3D assemblies, the full height of the sediments was investigated along the vertical direction, $z$, where the supernatant-sediment interface corresponds to $z = 0$ mm. The typical 2D µrad-SAXS patterns obtained at different heights in the sediment are presented in Fig. 4a for the soft fluorescent superballs with $L = 1.261 \pm 0.045$ µm and $m = 3.0$ dispersed in ethanol. At each height different patterns are recorded. Therefore, the $q$-positions of the $hk0$ peaks were determined and show that positions of the $q_{210}$ and $q_{110}$ peaks change differently as a function of $z$ indicating a continuous phase transition (Fig. 4b). Using the analysis described before we find that $\beta$ changes from 60° to 67.5° while $d$ decreases from $1.25L$ to $1.05L$ (Fig. 4c). Moreover, the visible $hkl$ peaks indicate that the PC and RC structures have dominant hollow-site stacking up to $z = -1.25$ mm followed by a small coexistence with bridge-site stacking until $z = -2.5$ mm, after which a full transition is observed from HRC to BRC, which was confirmed by rotation scans. Based on a structural angle of 60° and hollow-site stacking, the PC phase at the top corresponds to a face centred cubic structure. With increasing osmotic pressure this phase transforms into a HRC phase with a rhombic symmetry defined by the superball $m$ and, remarkably, followed

by a second crystal–crystal transition from a dominant HRC stacking into a BRC structure at the highest osmotic pressure. This soft superball system thus exhibits solid–solid transitions involving at least three distinct polymorphic structures, which is unique for colloids. For the other hard superball particles analysis of the µrad-SAXS patterns at different heights showed that similar PC to RC crystal–crystal transitions occur as a function of osmotic pressure (see Supplementary Fig. 6 and Supplementary Fig. 7). For the superballs with $m \leq 3.1$ this involves a PC–HRC phase transition while for the superballs with $m = 3.5$ this consist of a PC-BRC phase transition. For the superballs with $m = 3.6$ no transition is observed.

To properly compare the results for all superballs, which differ in size and dispersion medium, one has to take into account that the gravity effect scales with $L^4$. The heights were therefore rescaled by dividing by the gravitational length, $L_g$, which is a measure of the balance between the thermal energy and the gravitational force (See supplementary Table 1). Figure 4d shows $-z/L_g$ and the corresponding structural angle $\beta$. Note that the $-z/L_g$ scale is reverted such that the top of the graph corresponds to the sediment top. The PC to RC transition is clear as $\beta$ changes from $\sim 60°$ to the close-packed $\beta$ value corresponding to the particular superball $m$ at higher pressure. Based on the full height scan for each superball $m$ we can construct a phase diagram (Fig. 4e). It is clear that for the hard superball particles the crystal–crystal transition occurs at decreasing pressure for increasing $m$ and explains the absence of the transition for $m = 3.6$, where due to the $L_g/L$ ratio $\sim 1.4$ a very steep rise of the osmotic pressure occurs within the size of the X-ray beam. Interestingly, for the soft superballs with $m = 3.0$ and higher osmotic pressures ($L_g/L \sim 1.12$) two solid–solid transitions occur, with the PC–HRC transition at slightly higher pressures as the softer interactions make the particles more round. Together with the observation of the change in dominant stacking variant above $m \sim 3.2$ it is clear that the phase behaviour is controlled by the superball shape.

## Discussion

The experimentally observed phase behaviour in the superball sediments only confirmed part of the predicted phase behaviour of a transition from a PC to a rhombohedral crystal with $C_1$-lattice structure of superballs in the studied $m$-range[33,34]. At low osmotic pressure we indeed find the formation of a PC phase consisting of freely rotating particles in a truly long-range ordered face centred cubic-lattice, as shown by the combination of CSLM and µrad-SAXS experiments. In addition, for the superballs with $m \leq 3.1$ we find a transition into an HRC phase that has similar structural features as the close-packed $C_1$-lattice. The main difference between the structures is a slight rotation of the superball particles in the RC planes that we have considered flat throughout our analysis. In the µrad-SAXS patterns for $m \leq 3.1$ we cannot distinguish between the two superball orientations from $P(\mathbf{q})$. We cannot even detect the difference between the slightly different crystal structures from $S(\mathbf{q})$ since for both the structures the theoretical $S(\mathbf{q})$ will fall within the width of the experimentally observed $S(\mathbf{q})$. Therefore, we conclude that our findings present the first experimental evidence of the superball phase behaviour for $m \leq 3.1$.

The most surprising observation is the bridge-site stacking variant of the RC crystal for the hard superballs with $m \geq 3.5$ and even for the soft superballs with $m = 3.0$ at high pressures. This BRC phase was not predicted[33,34] and has not been observed for colloidal crystals before. The question remains whether the BRC structure is an equilibrium phase missed in the simulations or if it has been induced by experimental factors, such as wall-anchoring

or particle polydispersity in size or shape. Due to the high computing power required for free-energy calculations[34], predetermined optimal lattices[31] are often used as starting point, thereby excluding other crystal structures from the phase diagram. However, wall-anchoring effects can lead to a structural change perpendicular to the wall as shown by the coexistence of a small amount of BRC in the bulk HRC for superballs with $m \leq 3.1$. Further support comes from the observation that fixed simulation box boundaries induce an additional phase transition[33], however no structural details are provided preventing direct comparison. Furthermore, wall effects are not expected to range over several hundreds of crystal planes ($200 \, \mu m$). Another explanation could be an experimental deviation from the perfect hard superball shape, i.e., particle polydispersity or slightly flattened superball faces, which drives the system away from equilibrium or provides stability of lower density phases. For instance, it has been shown by simulations that the stability of the PC phase is very sensitive to the finite superball roundness[34], while for perfect cubes the alignment of the cube faces can stabilize crystal phases by delocalizing vacancies[44]. In addition, particle polydispersity, often seen as an enemy of the ordered state, has been found to stabilize unexpected or even exotic ordered phases for spheres and rods[45,46]. Clearly, further investigations both experimental and theoretical are needed to define which factors determine the experimental phase behaviour of the superballs studied here.

To summarize, the presented results show that in our experimental system of superballs, macroscopically large mono-crystalline structures can be achieved and that the stacking variant can be tuned by shape. Our findings open up ways to design materials that have potential for optical or other applications. Combining the knowledge about the shape with the extensive methods available to control the superball interactions could provide additional avenues for preparing new functional structures.

## Methods

**Colloidal superball synthesis.** Colloidal hollow superballs consisting of silica or of fluorescently labelled silica with different sizes and shapes were synthesized according to published literature protocols[30,38,40]. For the differently shaped and sized superballs we used different hematite superball precursor particles that were coated with a thin silica shell of $\sim 50$–$100 \, nm$ followed by removal of the hematite core. In order to distinguish particles in CLSM images some of the superballs consisted of a fluorescent core and outer non-fluorescent silica shell. The obtained superball particles were characterized by TEM (Phillips TECNAI 10/12). The specific properties of the superballs are given in Supplementary Table 1.

**Sample preparation.** Fluorescently labelled silica superballs were dispersed in ethanol (100%, Mercachem) and non-labelled silica superballs were dispersed in 6 mM tetramethyl ammonium hydroxide (TMAH, Fluka) aqueous solution with pH = 9. The alkaline nature of the solvents induces deprotonation of the silanol groups on the silica surface and hence provides charge stabilization. In the ethanol samples the Debye screening length is estimated as $\kappa^{-1} \sim 50 \, nm$. In the TMAH samples the high salt content screens the double layer repulsion and $\kappa^{-1}$ reduces to 4 nm.

**Confocal laser scanning microscopy.** CLSM measurements were performed on a Nikon TE2000U inverted microscope fitted with a C1 confocal scan head and a $100 \times$ Nikon oil objective using a HeNe laser (543.5 nm, Melles Griot) for excitation of the rhodamine dye. Samples were made in sedimentation cells consisting of a round glass capillary ($2 \times 100 \, mm$ internal dimension) glued with araldite glue to a glass cover slip of 0.17 mm thick (Menzel-Gläser). Superball dispersions were placed in the cells followed by flame sealing and the sedimentation was imaged over time in the first 15 µm above the glass wall.

**Microradian small angle X-ray scattering.** For µrad-SAXS measurements superball dispersions of 2–5 v% were placed in capillaries with internal dimensions of $100 \times 4 \times 0.2 \, mm$ (W3520 Vitrocom) or round capillaries with an internal diameter of 1 mm (Mark tubes) that were flame sealed and stored vertically.

Sedimentation occurred over a period of 24–72 h depending on the size of the superballs. Samples were measured within 1 week–1 month after sample preparation. μrad-SAXS measurements were performed at beam-line BM26B DUBBLE (refs 47–49) at the ESRF in Grenoble using a μrad-SAXS set-up employing compound refractive lenses[50–52]. The X-ray beam (13 keV) was focused 7.17 m behind the sample on the centre of a CCD X-ray detector with dimensions of 4,008 × 2,671 pixels and a pixel size of 9 × 9 μm (Photonic Science). The detector was protected from the direct X-ray beam using a wedge-shaped beam-stop that shades the detector. The modulus of the scattering vector is determined by the scattering angle $2\theta$ as $q=|\mathbf{q}|=4\pi \sin\theta/\lambda$. This set-up provided a range of $0.0022\,\mathrm{nm}^{-1} \leq q \leq 0.167\,\mathrm{nm}^{-1}$. Crystallinity of the samples was checked with white light illumination that showed the presence of distinct Bragg reflections. The capillaries were oriented vertically with their long axis (100 mm) parallel to the gravitational field and with their short axis (0.2 mm) parallel to the X-ray beam. The full height of the sediments was explored in detail by scanning with a step size (0.1 or 0.25 mm) smaller than the X-ray beam size on the sample (0.5 × 0.5 mm). At several height positions rotation scans were performed by rotating the sample around its vertical axis over a range of $\omega \pm 70°$ with a step size of 2.5° or 5°, where $\omega = 0°$ is the initial sample orientation. Dark-current and background corrections were performed on the patterns before analysis. For the background correction patterns were obtained of capillaries with pure solvent.

**Data availability.** The data that support the findings of this study are available from the corresponding authors on request.

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

## Acknowledgements

L. Rossi is thanked for many fruitful discussions. S.I.R. Castillo, V. Meester, F. Hagemans and L. Pompe are thanked for their contributions to the particle synthesis. M. van den Berg, K. van Berlo, R. van Dinther en J. Opdam are thanked for their contributions to the sedimentation experiments. W. Irvine is thanked for providing the MatLab scripts for analysis of the superball shape parameter *m*. J. van Rijssel is thanked for his help with the form factor calculations. A.B.G.M. Leferink op Reinink, J. Hilhorst and J.R. Wolters

are thanked for their help during the µrad-SAXS measurements. The whole DUBBLE team and especially G. Portale and D. Hermida Merino are thanked for their excellent technical support during the µrad-SAXS measurements. The Nederlandse Organisatie voor Wetenschappelijk Onderzoek (NWO) is acknowledged for the provided beam-time.

## Author contributions

J.-M.M. and A.V.P. designed the experiments. A.V.P., A.P.P. and H.N.W.L. supervised the project. J.-M.M. prepared samples and collected EM data. J.-M.M. A.P. and A.V.P. collected SAXS data and J.-M.M. and S.O. collected CLSM data. J.-M.M. analysed SAXS and CLSM data. J.-M.M. and A.V.P. wrote the manuscript. A.P., S.O., H.N.W.L. and A.P.P. edited the manuscript.

## Additional information

**Competing financial interests:** The authors declare no competing financial interests.

**Publisher's note**: 

