## [Peer Review File · Nature Communications]

Reviewers' comments:

Reviewer #1 (Remarks to the Author):

The manuscript investigates the crystallization of colloidal systems of nearly hard superballs realized as suspensions of micron-sized silica particles. The main discoveries are a plastic crystal with hard colloids and two rhombohedral phases, one of them not reported before. The main experimental tools are SAXS measurements.

Overall the findings appear sound, the analysis of high technical quality (especially the analysis of microrad-SAXS data is very well done), and the manuscript is well written. However, the key discoveries are expected, which limits the impact of the work. The newly reported crystal has only subtle differences to known phases (e.g., Zhang et al. PRL 107, 135701 (2011)) and the novelty is not clear given the known complexity of the theoretical phase diagram (Ni et al., Soft Matter 8, 8826 (2012)). Furthermore, plastic crystals have been observed in a few systems already (for example in the author's own work: Vutukuri et al., Ang. Chem. Int. Ed. 53, 13830 (2014), not cited yet) and this work adds to a growing list of such reports.

Detailed discussion:

(1) I find the observation of a plastic crystal not very surprising in this system. The authors write "a plastic crystal phase [...] was not reported earlier for hard colloids with short-range interactions only". While this might be technically true given the right definition for "short-range interactions", this statement gives a wrong impression of the novelty. Plastic crystals have now been found in various experiments using colloids, vibrated (hard) granular matter in 2D (arXiv:1608.07254), and nanoparticles.

(2) The "hollow-site" and "bridge-site" stacking variants can be interpreted as sheared versions of fcc and bcc, respectively. Hard spheres form fcc (or stacking variants) and thus sheared versions thereof are expected if the spherical shape is deformed cubically. Soft(er) spheres form bcc. Again, sheared versions are expected for cubic deformation. Thus, the observation of bridge-site stacking, which was not predicted in hard particle simulations, for superballs that have shapes deviating strongly from a sphere suggests to me that the colloids in this study possibly retain some degree of softness.

(3) Is the discrepancy between simulation and experiment a liability or an asset? The abstract states: "[...] assumptions implemented in the simulations based on geometrical constrains can sometimes influence the found phases." This is of course true, but also a trivial statement. It is also a negative view of the role of simulations. Instead of seeing the discrepancy as a problem, I suggest to interpret the discrepancy between simulation and experiment as an opportunity to learn something about the experiment. For example, the superballs in this work might not be perfectly hard after all. Softness could explain the findings (see above) and could be an alternative or complementary explanation to the authors' current hypothesis (that is: "wall anchoring").

(4) Three factors can drive the phase behavior away from equilibrium:

(i) Wall effects can be quite strong as seen in the dependence of the appearance of poly-crystallinity on capillary shape (Supplementary Figure 3). The rhombohedral lattice is often not sheared in a plane such that two of the cube faces are still aligned planar. But when deposited near a wall (the manuscript states: "in the rhombic crystal the superball faces were found to anchor to the wall") such a not-in-a-plane shear is strongly inhibited.

(ii) Another potential limitation to observe clean phase behavior is the fact that the experimental system is not homogeneous. Osmotic pressure increases from top to bottom, which leads to a gradual

phase change (Figure 4).

(iii) Finally, finite sedimentation speed might effect the phase behavior and lead to a deviation from equilibrium (e.g., the authors state: "the transition is not observed since the osmotic pressure raises too quickly").

All these effects make it difficult to interpret the findings and compare them to other (theoretical experimental) works.

(5) The distinction of the two stacking variants is quite difficult and the authors have done a great job detecting it in experiment. I find this distinction the most interesting part of the manuscript. In the cases when hollow-site stacking and bridge-site stacking are observed in coexistence, is it possible that one phase is templated by the wall and competes with the other, native (minimum free energy) phase?

(6) In which sense are the two rhombohedral lattice phases close-packed? Are they densest packings or are they simply packed in a way such that a local maximum in packing density is achieved which can be resolved only by relaxing packing density?

Reviewer #2 (Remarks to the Author):

This paper presents a sophisticated and careful experimental study of the phase behavior of silica superballs as a function of particle shape, osmotic pressure and, to a lesser extent, the particle-particle interaction. The most interesting results are the observation of two sequential solid-solid transitions for the 'softer' particles with $m = 3$, and the observation of a rhombohedral phase with bridge-site stacking in certain parameter regions. These results were not predicted by previous theoretical studies of hard superballs, and should be of strong interest to others in the field. The experimental observation of the theoretically predicted plastic crystal phase is also new.

This work is probably suitable for publication in Nature Communications, but the language and interpretation of results will need to be made more precise. For example:

- In the abstract, it is not when entirely clear when previous experimental and theoretical work is being referred to, which can lead to the false impression that the authors are claiming that the plastic crystal phase was not predicted by simulations.

- Two different types of colloidal suspensions are used, one with a longer Debye screening length resulting in softer interactions between particles and one with a shorter Debye screening length resulting in harder interactions between particles, however it is sometimes difficult to tell in a given paragraph which suspension is being discussed.

- The paper gives the impression that it is surprising that "assumptions implemented in the simulations based on geometrical constrains can sometimes influence the found phases". What would be surprising, is if particle shape and interactions did not influence phase behavior.

- The authors use vague language that attributes differences between the current experimental work and previous simulation studies of hard superballs to "assumptions implemented in the simulations" and "the influence of experimental conditions", without clearly specifying what these are. Do the authors think these differences are due to mistakes made in the previous simulation studies of hard superballs, differences in particle interactions, polydispersity of the particles used in the experiments (which appears to be considerable in the supplementary movie), the influence of surface-induced nucleation in the experiments, or something else?

- In the discussion the authors write "Therefore, we conclude that the bridge-site stacking sequence is the thermodynamically stable crystal structure for the experimental system of silica superballs", but it is unclear what precisely this conclusion is based on or what they think has stabilized this new structure.

- The main manuscript states that "The striking influence of the capillary shape on the long-range order is unexpected", but does not explain what the influence is or why it is unexpected. The reader is left to refer to Fig. S3 to discover that the "striking influence" is a transition from mono- to polycrystallinity as the capillary changes from having a rectangular to a circular cross-section, which is worth noting but hardly unexpected given that the nucleation appears to be surface-induced.

Reviewer #3 (Remarks to the Author):

The authors describe small angle x-ray scattering and confocal laser scanning microscopy experiments to elucidate the crystal phases formed from approximately 1 micron anisotropic colloidal silica particles. Specifically "superballs" (particles on the continuum between a sphere and a cube) with a shape parameter $m=2.9-3.6$ were studied.

They discovered that three phases were observable within the range of conditions studied (shape parameter and osmotic pressure). Namely a plastic crystal phase at lower osmotic pressures (long range translational order, but no orientation order), and two forms of rhombohedral crystal structures, differentiated by their stacking. "Bridge-site" stacking is favoured for higher values of the shape parameters (i.e., more anisotropic), compared to the "hollow-site" stacking.

The "bridge-site" stacking was not expected from simulation of these anisotropic systems - i.e., only two phases expected. The authors suggest these deviation from simulation could be due to anchoring at the walls of the capillary, or slight deviations of the shape of the real particles (i.e., flattened faces) from the simulated model.

A further interesting finding was that in flat sided capillaries very large crystals (essentially single crystal with respect to the x-ray beam) were formed, however, in cylindrical capillaries the sample was polycrystalline.

The experiments presented in this paper were well conceived, with the complimentary techniques of CLSM and SAXS providing sufficient information to characterise the system with confidence.

The paper is well written, with each section clearly explained, without being too verbose. The discussion and concluding remarks are well founded from the results presented.

Response to Reviewers:

First of all we would like to thank the reviewers very much for their constructive comments that helped us to clarify a number of key points in the manuscript. Due to the overlap in commentary we have decided that to incorporate all clarifications and changes properly, it was best to completely rewrite the abstract, introduction and discussion paragraphs. In the revised manuscript these sections and several sentences or words that were changed are marked in blue.

Below we provide further detailed point-by-point answers and explanations of the revisions made to the manuscript. The page numbers and paragraphs numbers refer to those of the revised manuscript.

Response to Reviewer 1:

We appreciate the overall comments of reviewer 1 in support of our work. We also agree with reviewer 1 that the main novelty of our work was not stated sufficiently clearly.

Changes in the manuscript: *To properly place the emphasis of this work, we have greatly re-written the abstract, introduction and discussion sections, and included the suggested references.*

Comment (1): I find the observation of a plastic crystal not very surprising in this system. The authors write "a plastic crystal phase [...] was not reported earlier for hard colloids with short-range interactions only". While this might be technically true given the right definition for "short-range interactions", this statement gives a wrong impression of the novelty. Plastic crystals have now been found in various experiments using colloids, vibrated (hard) granular matter in 2D (arXiv:1608.07254), and nanoparticles.

Answer: Reviewer 1 is right that plastic crystal phases have been observed before and that the phrasing in our manuscript might be misleading.

Changes in manuscript: *We have addressed these issues by rewriting the introduction paragraphs and removing the statement about the novelty of the observation a plastic crystal phase.*

Comment (2): The "hollow-site" and "bridge-site" stacking variants can be interpreted as sheared versions of fcc and bcc, respectively. Hard spheres form fcc (or stacking variants) and thus sheared versions thereof are expected if the spherical shape is deformed cubically. Soft(er) spheres form bcc. Again, sheared versions are expected for cubic deformation. Thus, the observation of bridge-site stacking, which was not predicted in hard particle simulations, for superballs that have shapes deviating strongly from a sphere suggests to me that the colloids in this study possibly retain some degree of softness.

Answer: We thank Reviewer 1 for this interesting analogy. For spheres the "bridge-site" stacking leads to a body-centred tetragonal (BCT) structure, which is similar to BCC but "compressed" along one of the cubic axes. BCT is typical for dipolar spheres [Pal, A., Malik, V., He, L., Ern , B. H., Yin, Y., Kegel, W. K. and Petukhov, A. V. (2015), *Angew. Chem. Int. Ed.*, 54: 1803–1807, Yethiraj and van Blaaderen, (2003) *Nature* 421, 513-517]. We do not think that the transition from hollow- to bridge-site stacking is related to any kind of "softness" in interactions between them. In our experiments the Debye length was varied from 4 nm (=very hard particles) to 50 nm (=with slight softness). The bridge-site stacking structure was only observed for very hard particles while the systems with some repulsion always showed hollow-site stacking. This is exactly opposite to the suggestion of the referee. We instead relate the formation of the structure to the exact shape of the superball, as the bridge-site staking is observed for superballs with the highest m -values. The comment of reviewer 1 does show that our conclusion was not made sufficiently clear.

Changes in manuscript: *We have clearly stated the ‘softer’ or ‘hard’ nature of the cubes and adjusted the discussion about the reason why a bridge-site stacking structure is found in this experimental system of superball particles.*

Comment (3): Is the discrepancy between simulation and experiment a liability or an asset? The abstract states: "[...] assumptions implemented in the simulations based on geometrical constraints can sometimes influence the found phases." This is of course true, but also a trivial statement. It is also a negative view of the role of simulations. Instead of seeing the discrepancy as a problem, I suggest to interpret the discrepancy between simulation and experiment as an opportunity to learn something about the experiment. For example, the superballs in this work might not be perfectly hard after all. Softness could explain the findings (see above) and could be an alternative or complementary explanation to the authors' current hypothesis (that is: "wall anchoring").

Answer: The referee is absolutely right that any difference between theory and experiment motivates further studies from both sides. We also agree that we stressed too much the discrepancy issue. However, as explained above, we do not think that softness is the cause of the deviations. An alternative is that experimentally the particles are not perfectly superball shaped but possess slightly flatter faces, which can cause anchoring of the particles. We have taken these points into account during rewriting of the discussion section.

Changes in manuscript: *The mentioned statement about the simulations has been removed from the abstract and the discussion has been rewritten to remove the seemingly negative view of the simulations.*

Comment (4): Three factors can drive the phase behavior away from equilibrium:

(i) Wall effects can be quite strong as seen in the dependence of the appearance of poly-crystallinity on capillary shape (Supplementary Figure 3). The rhombohedral lattice is often not sheared in a plane such that two of the cube faces are still aligned planar. But when deposited near a wall (the manuscript states: "in the rhombic crystal the superball faces were found to anchor to the wall") such a not-in-a-plane shear is strongly inhibited.

Answer: This is indeed an important point. A flat wall can promote “flattening” of the crystal planes (and in these studied systems the walls do serve as a nucleation point). However, the opposite walls of the flat capillaries used in this study are separated by 200 microns, i.e. several hundred of crystal planes. One can expect the anchoring effect in the close vicinity of a flat wall but it is difficult to imagine that it will penetrate deep into the bulk as the transition between two types of structures can be easily achieved by a tiny rotation of the particles. Exactly this effect is observed for the slightly rounder superballs where a small amount of bridge site stacking is present but the bulk shows the hollow site stacking variant (as also realized by Reviewer 1 in comment (5)). Based on this coexistence behavior we believe that the phase behavior observed for the more cubic superballs is the equilibrium state of the experimental system. Why this state forms is still unclear to us. From this comment of reviewer 1 it is clear that this point was not sufficiently clearly stated in our manuscript.

Changes in manuscript: *In the discussion the wall-anchoring is addressed along with the consequences of this effect.*

Comment (4) (ii) Another potential limitation to observe clean phase behavior is the fact that the experimental system is not homogeneous. Osmotic pressure increases from top to bottom, which leads to a gradual phase change (Figure 4).

Answer: Yes, there is a gradient of the osmotic pressure leading to a gradual transition from one phase to another. It happens, however, on the scale of a millimeter in our samples, which is about thousand particles sizes. We therefore believe that the pressure changes adiabatically and the local unit cell

structure is able to follow the gradually changing pressure, which is also seen in the gradual change of the structural angle.

Changes in manuscript: *No direct changes were made in the manuscript.*

Comment (4) (iii) Finally, finite sedimentation speed might effect the phase behavior and lead to a deviation from equilibrium (e.g., the authors state: "the transition is not observed since the osmotic pressure raises too quickly"). All these effects make it difficult to interpret the findings and compare them to other (theoretical experimental) works.

Answer: Yes, there are obviously a number of experimental details that can also affect the structures observed. Apart from further experimental studies, we hope that additional simulations can help to clarify the origin of the phases observed here.

Changes in manuscript: *We have included a statement in the discussion about the difficulty of comparing the found phases to the simulation studies previously performed.*

Page 8. 3rd paragraph. *"The question remains whether the BRC structure is an equilibrium phase missed in the simulations or if it has been induced by experimental factors, such as wall-anchoring or particle polydispersity in size or shape."*

Comment (5A): The distinction of the two stacking variants is quite difficult and the authors have done a great job detecting it in experiment. I find this distinction the most interesting part of the manuscript.

Answer: We thank reviewer 1 for the kind words about our experimental findings. We agree that the distinction is indeed a very interesting point and as stated before we have changed the introduction and discussion paragraphs to reflect this (see also answers above).

Comment (5B): In the cases when hollow-site stacking and bridge-site stacking are observed in coexistence, is it possible that one phase is templated by the wall and competes with the other, native (minimum free energy) phase?

Answer: This is a good point of Reviewer 1 and agrees with our interpretation of the results. Apparently, we did not express our conclusion clearly in the discussion.

Changes in manuscript: *We have added a clear statement in the discussion about the coexistence of the two phases.*

Page 8. 3rd paragraph: *"However, wall-anchoring effects can lead to a structural change perpendicular to the wall as shown by the coexistence of a small amount of BRC in the bulk HRC for superballs with $m \leq 3.1$."*

Comment (6): In which sense are the two rhombohedral lattice phases close-packed? Are they densest packings or are they simply packed in a way such that a local maximum in packing density is achieved which can be resolved to only be relaxing packing density?

Answer: This is a good question of Reviewer 1. As we find that the hollow site stacking is similar to C_1 -lattice, this structure provides the optimal packing for the superballs and hence the closest packing. The bridge site stacking due to the orientation of the cubes, has to be slightly lower in density. However, as mentioned by Reviewer 1 as well, this orientation could allow in-plane shear that stabilizes the lower density phase.

Changes in manuscript: *We included in the discussion a clear statement about the close-packed nature of the hollow-site RC lattice and the fact that it is similar to the close-packed C_1 -lattice.*

Response to Reviewer 2:

We appreciate the nice comments of reviewer 2 about our manuscript. We thank Reviewer 2 for pointing out the unclear statements in our manuscript.

Comment: In the abstract, it is not when entirely clear when previous experimental and theoretical work is being referred to, which can lead to the false impression that the authors are claiming that the plastic crystal phase was not predicted by simulations.

Answer: This is indeed an important point and we rewritten the abstract to clarify the difference between experiment and simulations.

Changes in manuscript: *The abstract has been rewritten.*

Comment: Two different types of colloidal suspensions are used, one with a longer Debye screening length resulting in softer interactions between particles and one with a shorter Debye screening length resulting in harder interactions between particles, however it is sometimes difficult to tell in a given paragraph which suspension is being discussed.

Answer: We thank Reviewer 2 for pointing out this difficulty and rephrased several sentences in the manuscript to clarify the system that is being discussed.

Changes in manuscript: *Rephrased the statements concerning the different particles.*

Page 3. 3rd paragraph: *“The Debye screening length was estimated to be $\kappa^{-1} \sim 50$ nm leading to slightly ‘soft’ interactions...”*

Page 4. 3rd paragraph: *“[...] of the ‘softer’ fluorescent silica superballs with μ rad-SAXS. In addition, to see the effect truly ‘hard’ interactions as well as the effect of shape, we also studied the sediments of unlabelled silica superballs with different m values.”*

Page 6. 2nd paragraph: *“Noteworthy is that both the ‘softer’ fluorescent superballs with $m = 3.0$ and the differently shaped ‘hard’ superballs all have formed close packed structures.”*

Page 7. 1st paragraph:

“[...] Fig.4a for the ‘softer’ fluorescent superballs [...]”

“This ‘softer’ superball system thus exhibits solid-solid transitions involving at least three distinct polymorphic structures, which is unique for colloids. For the other ‘hard’ superball particles analysis [...]”

Page 8. 3rd paragraph: *“ [...]for the ‘hard’ superballs with $m \geq 3.5$ and even for the ‘softer’ superballs with $m = 3.0$ [...].”*

Comment: The paper gives the impression that it is surprising that "assumptions implemented in the simulations based on geometrical constrains can sometimes influence the found phases". What would be surprising, is if particle shape and interactions did not influence phase behavior.

Answer: We agree with Reviewer 2 that this statement was confusing. Therefore it was removed from the introductory paragraphs.

Changes in manuscript: *Besides removing this statement we have eliminated other ambiguous statements from the manuscript, by rewriting the abstract, introduction and discussion.*

Comment: The authors use vague language that attributes differences between the current experimental work and previous simulation studies of hard superballs to "assumptions implemented in the simulations" and "the influence of experimental conditions", without clearly specifying what these are. Do the authors think these differences are due to mistakes made in the previous simulation studies

of hard superballs, differences in particle interactions, polydispersity of the particles used in the experiments (which appears to be considerable in the supplementary movie), the influence of surface-induced nucleation in the experiments, or something else?

Answer: We agree with Reviewer 2 that these statements require further explanation. We think that there are several factors that can cause the differences between the experiments and simulations and have rewritten the discussion such that the different causes are discussed clearly.

Changes in manuscript: *“The most surprising observation is the bridge-site stacking variant of the RC crystal for the ‘hard’ superballs with $m \geq 3.5$ and even for the ‘softer’ superballs with $m = 3.0$ at high pressures. This BRC phase was not predicted^{33,34} and has not been observed for colloidal crystals before. The question remains whether the BRC structure is an equilibrium phase missed in the simulations or if it has been induced by experimental factors, such as wall-anchoring or particle polydispersity in size or shape. Due to the high computing power required for free-energy calculations³⁴, predetermined optimal lattices³¹ are used as starting point, thereby excluding other crystal structures from the phase diagram. However, wall-anchoring effects can lead to a structural change perpendicular to the wall as shown by the coexistence of a small amount of BRC in the bulk HRC for superballs with $m \leq 3.1$. Further support comes from the observation that fixed simulation box walls induce an additional phase transition³³, however no structural details are provided preventing direct comparison. Furthermore, wall effects are not expected to range over several hundred of crystal planes (200 μm). Another explanation could be an experimental deviation from the perfect ‘hard’ superball shape, i.e. particle polydispersity or slightly flattened superball faces, which drives the system away from equilibrium or provides stability of lower density phases. For instance, it has been shown by simulations that the stability of the PC phase is very sensitive to the finite superball roundness³⁴, while for perfect cubes the alignment of the cube faces can stabilize crystal phases by delocalizing vacancies⁴⁴. In addition, particle polydispersity, often seen as an enemy of the ordered state, has been found to stabilize unexpected or even exotic ordered phases for spheres and rods^{45,46}. Clearly, further investigations both experimental and theoretical are needed to define which factors determine the experimental phase behaviour of the superballs studied here.”*

Comment: In the discussion the authors write "Therefore, we conclude that the bridge-site stacking sequence is the thermodynamically stable crystal structure for the experimental system of silica superballs", but it is unclear what precisely this conclusion is based on or what they think has stabilized this new structure.

Answer: We again agree with Reviewer 2 that further explanation is needed and have taken this into account in the rewriting of the discussion paragraphs.

Changes in manuscript: *As stated above the discussion paragraph has been rewritten.*

Comment: The main manuscript states that "The striking influence of the capillary shape on the long-range order is unexpected", but does not explain what the influence is or why it is unexpected. The reader is left to refer to Fig. S3 to discover that the "striking influence" is a transition from mono- to poly-crystallinity as the capillary changes from having a rectangular to a circular cross-section, which is worth noting but hardly unexpected given that the nucleation appears to be surface-induced.

Answer: Yes, the influence of the capillary shape in a surface-induced nucleation is not striking. What is striking is that a mono-crystalline structure is formed over a range of ~ 500 particle diameters due to the presence of the wall.

Changes in manuscript: *We have changed the statements in the manuscript and revised the order of the paragraph.*

Page 5. 2nd paragraph: *“The formation of mono-crystalline domains, which are larger than the diameter of x-ray beam $\sim 500 \mu\text{m}$, is attributed to alignment of the superball faces to the capillary wall.*

This is followed by subsequent nucleation of the crystals consisting of close-packed RC planes. The wall-anchoring was confirmed by measurements in round capillaries where poly-crystalline patterns were observed (Supplementary Fig.3)."

Response to Reviewer 3:

We appreciate the nice comments of Reviewer 3 about our manuscript and thank Reviewer 3 for the support.

Other changes in the manuscript

We choose to abbreviate the different stacking sequences as we feel it improves the readability of the manuscript:

Plastic crystal – PC

Rhombohedral crystal – RC

Rhombohedral crystal with hollow-site stacking – HRC

Rhombohedral crystal with bridge-site stacking – BRC

REVIEWERS' COMMENTS:

Reviewer #1 (Remarks to the Author):

The authors have done a good job implementing my suggestions and updating the description and interpretation of their findings. At this point I agree with publication in Nature Communications without the need of further revisions.

Reviewer #2 (Remarks to the Author):

The authors have addressed all comments to the satisfaction of this reviewer.